# TempEL: Linking Dynamically Evolving and Newly Emerging Entities

**Klim Zaporojets**[♭]    **Lucie-Aimée Kaffee**[♯]    **Johannes Deleu**[♭]
**Thomas Demeester**[♭]    **Chris Develder**[♭]    **Isabelle Augenstein**[♯]
[♭] Ghent University – imec, IDLab, Ghent, Belgium
[♯] Dept. of Computer Science, University of Copenhagen, Denmark
`{klim.zaporojets,johannes.deleu,thomas.demeester,chris.develder}@ugent.be`
`{kaffee,augenstein}@di.ku.dk`

## Abstract

In our continuously evolving world, entities change over time and new, previously non-existing or unknown, entities appear. We study how this evolutionary scenario impacts the performance on a well established *entity linking* (EL) task. For that study, we introduce TempEL, an entity linking dataset that consists of time-stratified English Wikipedia snapshots from 2013 to 2022, from which we collect both *anchor mentions* of entities, and these *target entities*' descriptions. By capturing such temporal aspects, our newly introduced TempEL resource contrasts with currently existing entity linking datasets, which are composed of fixed mentions linked to a single static version of a target Knowledge Base (e.g., Wikipedia 2010 for CoNLL-AIDA). Indeed, for each of our collected temporal snapshots, TempEL contains links to entities that are *continual*, i.e., occur in all of the years, as well as completely *new* entities that appear for the first time at some point. Thus, we enable to quantify the performance of current state-of-the-art EL models for: (i) entities that are subject to changes over time in their Knowledge Base descriptions as well as their mentions' contexts, and (ii) newly created entities that were previously non-existing (e.g., at the time the EL model was trained). Our experimental results show that in terms of temporal performance degradation, (i) *continual* entities suffer a decrease of up to 3.1% EL accuracy, while (ii) for *new* entities this accuracy drop is up to 17.9%. This highlights the challenge of the introduced TempEL dataset and opens new research prospects in the area of time-evolving entity disambiguation.[1]

## 1 Introduction

Entity linking (EL) is a well-established task that is concerned with mapping anchor *mentions* in text to target *entities* that describe them in a Knowledge Base (KB) (e.g., Wikipedia).[2] Existing benchmark datasets for EL [73, 66, 71, 57] are composed of a fixed set of annotated mentions linked to a single version of a target KB. This static setup is oblivious to the inherently non-stationary nature of the entity linking task where both target entities as well as anchor mentions change over time. The example in Fig. 1 illustrates this time-evolving essence of entity linking with a simple evolutionary comparison between Wikipedia 2013 and 2022. It showcases two scenarios studied in the current paper: (i) temporal evolution of existing (*continual*) entities across temporal snapshots,

---

[1]TempEL dataset, code and models are made public at `https://github.com/klimzaporojets/TempEL`.

[2]Some of the related work [19, 37, 71, 88, 86] distinguishes between *entity disambiguation* and *entity linking* tasks. This latter including *mention detection* and *disambiguation* in an end-to-end setting. In the current work, we follow a more conservative naming convention [61, 78, 43, 53, 60], and use the term *entity linking* and *entity disambiguation* interchangeably.

36th Conference on Neural Information Processing Systems (NeurIPS 2022) Track on Datasets and Benchmarks.

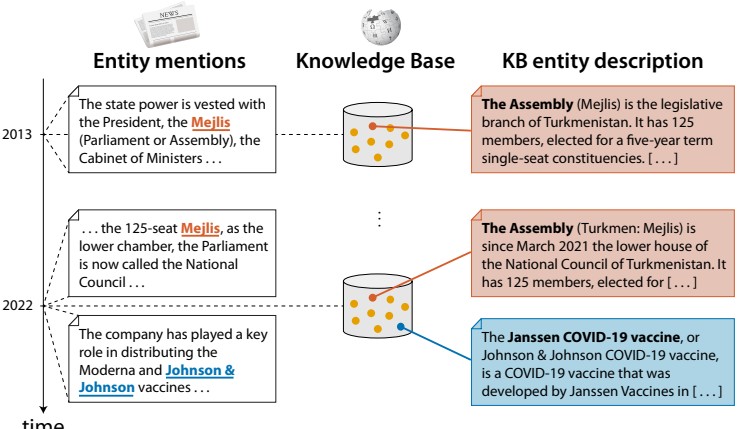

Figure 1: Illustration of KB entities changing over time: the "Mejlis" entity changes over time (both in its KB description and the contexts in which it is referenced to), while the Johnson & Johnson vaccine is an entirely new one that did not exist before.

and (ii) appearance of *new*, previously non-existent entities. For instance, the description of the *continual* entity *The Assembly* differs between Wikipedia 2013 and 2022. Furthermore, the context of a mention "Mejlis" referring to *The Assembly* also changes over time. Conversely, the *new* entity *Janssen COVID-19 vaccine* is newly introduced in 2021 with the corresponding mentions (e.g., "Johnson & Johnson" in Fig. 1) that are linked to it.

In this paper we introduce TempEL, a novel dataset to study this time-evolving aspect of the entity linking task. We therefore extract 10 equally spread yearly snapshots from English Wikipedia entities starting from January 1, 2013 until January 1, 2022. We use each of these temporal snapshots of Wikipedia to also extract anchor mentions with the surrounding text. Thus, TempEL captures the temporal evolution not only in the target entities as they are defined in the Wikipedia KB, but also in the contexts of anchor mentions linked to these entities. Each of the 10 temporal snapshots of our dataset is composed of training, test and validation sets with equal numbers of mentions and entities across the snapshots. Furthermore, TempEL is designed to comprise mentions pointing to *continual* entities across all the temporal snapshots, and to *new* entities inside a given temporal snapshot.

Finally, as a baseline, we finetune and evaluate the bi-encoder component of the BLINK model [78] on the various temporal snapshots of our newly introduced TempEL dataset. The bi-encoder is widely used in state-of-the-art entity linking models [88, 78] to retrieve the top $K$ (in this work we experiment with $K = 64$) candidate target entities for a given anchor mention context. Furthermore, its straightforward finetuning and fast retrieval performance on millions of candidate entities [32], make it an ideal choice to test on TempEL. Our experiments demonstrate a consistent temporal model deterioration for mentions linked to both *continual* (3.1% accuracy@64 points) as well as *new* (17.9% accuracy@64 points) entities. A more detailed analysis reveals that the maximum drop in performance is observed for *new* entities that require fundamentally different world knowledge that was not present in the corpus originally used to pre-train BERT. This is e.g. the case for *new* entities related to COVID-19 for which the bi-encoder model suffers additional deterioration of 14% accuracy@64 points compared to the rest of the new entities.

## 2 Related work

Our work is related to multiple different, yet interconnected research areas described below. First, we explain how TempEL compares to the currently widely used *entity linking datasets*. Next, we relate our work to already existing *temporal datasets* covering different aspects of the temporal evolution of the data. Finally, we describe the existing *entity-centric* research efforts, comparing the TempEL entity linking dataset to other datasets that heavily depend on the use of entities.

**Entity linking datasets** Most current state-of-the-art EL models [81, 55, 10, 88, 9] report on datasets from predominantly the news domain such as AIDA [25], KORE50 [25], AQUAINT [48], ACE 2004, MSNBC [62], N$^3$ [65], DWIE[87], VoxEL[68], and TAC-KBP 2010-2015 [28, 29]. Other frequently used datasets include the web-based IITB [38] and OKE 15/16 [51], as well as the tweet-based Derczynski [13]. Additionally, larger yet automatically annotated datasets such as WNED-WIKI and WNED-CWEB [20] have been also widely adopted. Finally, a number of resources such as the domain-specific biomedical MedMentions [49], the zero-shot ZeShEL [43], and the multi tasking DWIE [87] and AIDA$^+$[86] datasets have been recently introduced. Many of the mentioned datasets are further covered by entity linking evaluation frameworks such as GERBIL [73, 66] and KILT [57] that provide a common interface to evaluate the models. Yet, the mentioned resources are limited to static mention annotations linked to entities from a single version of a Knowledge Base. This contrasts with our newly introduced TempEL dataset, where the anchor mentions as well as the target entity descriptions are taken from different time periods. The datasets most closely related to our work are the recently introduced WikilinksNED [17, 53] and ShadowLink [59]. WikilinksNED contains only unseen mention-entity pairs in its test subset, thus encouraging the design of models invariant to overfitting and memorization biases. Furthermore, ShadowLink contains *overshadowed entities*: entities referred to by ambiguous mentions whose most likely target entity is different, e.g., the anchor mention "Michael Jordan" linked to the scientist instead of to the more widely referred to target entity describing the former basketball player. We incorporate the challenges presented in both of these datasets in TempEL (see Section 3.1 for further details).

**Temporal datasets** Research on temporal drift in data has gained a lot of interest in recent years. The focus has mostly been on creating datasets to train language models on different temporal snapshots of corpora derived from scientific [39], newswire [39, 15], Wikipedia [27], and Twitter [44] domains. More recently, temporal datasets have appeared to address tasks such as sentiment analysis [45, 50, 2], text classification [26, 22], named entity recognition [12, 64], question answering [39], and entity typing [46], among others. However, the creation of datasets tackling the temporal aspect of entity linking has largely been left unexplored. To the best of our knowledge, the dataset most closely related to TempEL is diaNED, introduced by [3]. There, the authors annotate mentions that require additional temporal information from the context to be correctly disambiguated. Conversely, in TempEL both mentions and entities are extracted from evolving temporal snapshots.

**Entity-driven datasets** Recent research has demonstrated the benefits of incorporating entity knowledge in various downstream tasks [82, 56, 79, 21, 74, 84, 42]. This progress has been accompanied by the creation of entity-driven datasets for tasks such as language modeling [58, 1, 36], question answering [85, 33, 31, 41, 70], fact checking [72, 54, 4] and information extraction [83, 87], to name a few. Yet, recent findings [69, 18, 40, 75, 23, 63] suggest that entity *representation* and *identification* (i.e., identifying the correct entity that match a given text) are among the main challenges that should be solved to further increase performance on such datasets. We believe that TempEL can contribute to addressing these challenges by: (i) encouraging research on devising more robust methods to creating *entity representations* that are invariant to temporal changes; and (ii) improving entity identification for non-trivial scenarios involving ambiguous and uncommon mentions (e.g., linked to *overshadowed entities* as defined above).

## 3 The TempEL dataset

In this section we will provide details on how TempEL was constructed (Section 3.1), describing the main components of the creation pipeline as sketched in Fig. 2. Furthermore, we discuss the aspects taken into account to guarantee the overall quality of our dataset (Section 3.2). Finally, we present statistics of TempEL (Section 3.3), illustrating its dynamically evolving nature.

### 3.1 Dataset construction

**Snapshot Data Extraction** As Fig. 2 indicates, we start from the history log dumps from February 1, 2022 of Wikipedia itself. We first filter these (see *Entity Filter* in Fig. 2) to: (i) exclude pages that are irrelevant for TempEL (i.e., categories, disambiguation pages, redirects and lists); and (ii) select the most temporally stable version of a Wikipedia page from the last month of the year in order to avoid introducing more volatile and potentially corrupted content edits (see Section 3.2 for further

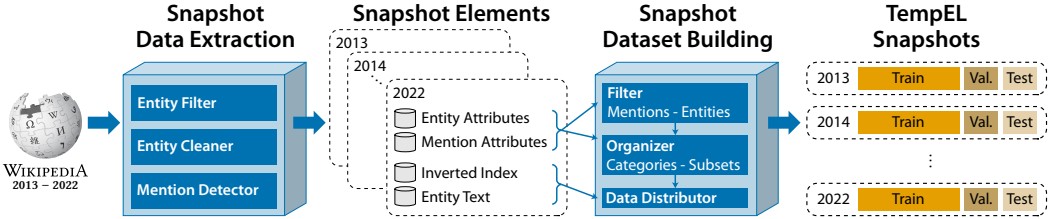

Figure 2: The pipeline to create our TempEL dataset. All the components are explained in Section 3.1.

details). Next, the Wikipedia pages are cleaned (see *Entity Cleaner* in Fig. 2) by stripping from the Wikitext markup content.[3] We use both regular expressions as well as the MediaWiki API for more difficult cases, such as the parsing of some of the Wikitext templates. Finally, we detect the mentions (see *Mention Detector* in Fig. 2) in each of the Wikipedia entity pages, filtering out the ones that point to anchors (i.e., subsections in Wikipedia pages), pages in languages other than English, files, red links (i.e., links pointing to not yet existing Wikipedia pages) and redirects.

The output of the *Snapshot Data Extraction* step first of all includes a set of *Entity* and *Mention Attributes* (e.g., the last modification date of the target entity), which are detailed in the supplementary material (see Section A.6). These attributes form part of the final dataset, making it possible to perform additional analyses of the results. Furthermore, the *Inverted Index* is generated to quickly access the Wikipedia pages that include a mention for a given target entity. Finally, the *Entity Text* files are extracted containing the (potentially yearly varying) textual content from the Wikipedia entity definition, as well as anchor mentions therein. These mentions of Wikipedia anchors that link to an entity will be extracted in the *Snapshot Dataset Building* step described further.

**Snapshot Dataset Building**   Starting from the *Snapshot Elements* produced by the *Snapshot Data Extraction* process described above, the actual TempEL dataset is now generated. The first step is to apply an additional *Filter* to both entities and mentions with the goal of creating a more challenging dataset. This is done by excluding mentions for which the correct entity it refers to has the highest prior [80]. More formally, the *mention prior* is calculated as follows,

$$P(e|m) = |A_{e,m}|/|A_{*,m}|, \tag{1}$$

where $A_{*,m}$ is the set of all anchors that have the same mention $m$, and $A_{e,m}$ is the subset thereof that links to entity $e$. Additionally, we exclude the mentions whose normalized edit distance from the target entity title is below an established threshold.[4] By ignoring the mentions with the highest prior and exact match with the title, we ensure that TempEL contains non-trivial disambiguation cases where the naive approaches (e.g., defaulting to the most frequently linked entity for a given mention) would fail [20, 43, 78, 59].

Furthermore, the entities are organized (see *Organizer* in Fig. 2) into two *categories*: (i) *new*, emerging and previously non-existent entities that are introduced in a particular snapshot; and (ii) *continual* entities across all the temporal snapshots. Next, the mentions are divided in separate subsets (i.e., train, validation and test), with the constraint of normalized edit distance between the mentions in different subsets referring to the same target entity be higher than 0.2. This way, we expect to discourage potential models from memorizing the mapping between mentions and entities [53].

Finally, the data is distributed equally (see *Data Distributor* in Fig. 2) across all of the temporal snapshots. This way, the difference in performance can only be attributed to temporal evolution and not to inconsistencies related to dataset variability (e.g., different number of training instances in each of the temporal snapshots). Concretely, we enforce that the number of *continual* and *new* entities as well as the number of mentions stays the same across the temporal snapshots (see Table 1). We achieve this by performing a random mention subsampling in snapshots with higher number of mentions, weighted by the difference in the number of mentions-per-entity. This produces a very similar mention-entity distribution across the temporal snapshots. Finally, the filtered anchor mentions are located in the cleaned Wikipedia pages (i.e., the *Entity Text* in Fig. 2) using the *Inverted Index* created in the previous *Snapshot Data Extraction* step. The context of each of the mentions

---

[3]`https://en.wikipedia.org/wiki/Help:Wikitext`
[4]During the generation of TempEL, we use a threshold of 0.2.

Table 1: Summary statistics of TempEL. The number of entities and mentions is the same across all of the temporal snapshots.

| Statistic | Train | Validation | Test |
|---|---|---|---|
| Temporal Snapshots | 10 | 10 | 10 |
| Continual Entities | 10,000 | 10,000 | 10,000 |
| # Anchor Mentions | 136,227 | 42,096 | 46,765 |
| New Entities | 373 | 373 | 373 |
| # Anchor Mentions | 1,764 | 1,231 | 1,450 |

is further paired with the respective content of target pages, outputting this way the final TempEL dataset.

## 3.2 Quality control

**Corrupted content**   Wikipedia is an open resource that relies on efforts of millions of Wikipedians to update and extend its contents.[5]   As such, that content is not always reliable, with errors due to human mistakes or intentional vandalism. Despite efforts to prevent the introduction of such erroneous edits [77, 8, 76], we have detected numerous cases of corrupted entity descriptions during our preliminary tests. As a result, we adopted a simple, yet very effective heuristic: for each of the entities of a particular yearly snapshot, we select the most *stable* (i.e., the version of the entity that lasted the longest before being changed) content of the last month of the year (December). Due to the fact that most of the corrupted content is rolled back very quickly, and even automatically by specialized bots [89, 30], this heuristic is very robust. We double checked the correctness of the extracted content by manually inspecting the evolution of hundred entities with lowest Jaccard vocabulary similarity between temporal snapshots and observed no obviously erroneous entries.

**Entity relevance**   We filter out entities that have less than 10 in-links (i.e., number of mentions linking to the entity) or contain less than 10 tokens in its Wikipedia page in order to avoid including noisy content [17]. Additionally, in order to avoid evaluation bias towards mentions pointing to more popular entities [55, 7], we limit the number of mentions per entity to 10 for our test and validation sets. This way, we expect the accuracy scores to not be dominated by links to popular target entities (i.e., entities with a big number of incoming links).

**Content filtering**   We only consider mentions linked to the main Wikipedia articles describing entities. The mentions pointing to anchors (subsections in a Wikipedia document), images, files, and wiki pages in other languages are filtered out in *Snapshot Data Extraction* step (see Fig. 2). In this step we also ignore pages that are not Wikipedia articles (e.g., files, information on Wikipedia users, etc.) as well as redirect pages. This way, the target entities as well as anchor mentions in our dataset are obtained from a cleaned list of candidate pages referring to entities that contain a meaningful textual description in Wikipedia.

**Dataset distribution**   During the construction of TempEL, we constrain the subsets to be of equal size and contain similar mention-per-entity distributions across all the temporal snapshots. This is implemented in *Data Distributor* sub-component of the dataset creation pipeline (see Section 3.1). For example, the number of mentions linked to continual entities in our training subset is 136,227 across all of the snapshots (see Table 1 for further details). We argue that this setting will produce uniform, structurally unbiased snapshots. This will allow to study exclusively the temporal effect on the performance of the models for each of the different time periods. Our reasoning is supported by previous work demonstrating that the size alone of the training set [44] as well as a different distribution of the number of mentions per entity [55] can significantly affect the performance of the final model. Furthermore, we do not constrain the total number of entities from the Wikipedia KB to be equal across the temporal snapshots (see Fig. 4c), since we consider it a part of the evolutionary nature of the entity linking task (i.e., the temporal evolution of the target KB) we intend to study.

---

[5]`https://en.wikipedia.org/wiki/Wikipedia:Wikipedians`

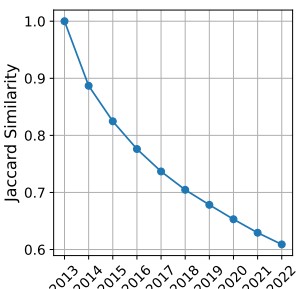 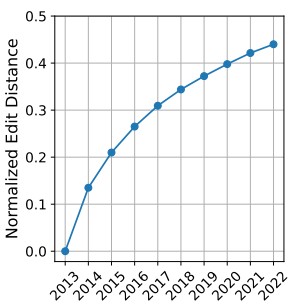 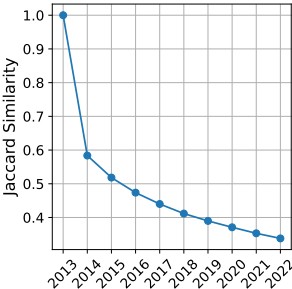

(a) Evolution of entities in terms of Jaccard vocabulary similarity.

(b) Evolution of entities in terms of edit distance of the content.

(c) Evolution of context around the mentions (Jaccard similarity).

Figure 3: Change of textual content of entities and context around mentions across temporal yearly snapshots (x-axis).

**Flexibility and extensibility**    Finally, we provide a framework that can be used to re-generate the dataset with different parameters as well as to extend it with newer temporal snapshots. This includes the option to generate a new dataset with a customized number of temporal snapshots (e.g., quarterly instead of yearly spaced), different mention attributes (e.g., filtering by mention prior values), entity popularity (e.g., filtering out entities that have more than a certain number of in-links), among others (see Section A.4 of the supplementary material for a complete list).

### 3.3  Dataset statistics

Table 1 summarizes the dataset statistics. We divide each of the temporal snapshots into train, validation and test subsets containing an equal number of *continual* and *new* entities. The number of mentions differs between the subsets since we limit the number of mentions per entity to 10 in both validation and test sets (see *entity relevance* in Section 3.2 for further details).

Additionally, we collect statistics related to temporal drift in content for both the target entities (Figs. 3a and 3b) as well as the context around the anchor mentions (Fig. 3c). Concretely, Fig. 3a visualizes Jaccard vocabulary similarity between the textual description of *continual* entities in 2013 and that of posterior yearly snapshots in TempEL. We observe a continual decrease, indicating that on average, the content of the entity description in Wikipedia is constantly evolving in terms of the used vocabulary. This is also supported by the graph in Fig. 3b, which showcases a continuous temporal increase of the average value of normalized edit distance across *continual* entities. Finally, Fig. 3c illustrates the temporal drift in the vocabulary (i.e., Jaccard vocabulary similarity) of the context around the mentions pointing to the same entity. We find it experiences a more significant change compared to the Jaccard similarity of entity content illustrated in Fig. 3a. This suggests that the context around the anchor mentions is subject to a higher degree of temporal transformation compared to that of target entities, making it an interesting item of future work.

## 4  Experiments

Our final TempEL comprises 10 different yearly snapshots and we evaluate entity linking (EL) performance on each of them individually. This evaluation setup allows us to study the effect of temporal corpus changes and assess the impact of increasing time lapses between the data used for model training and that on which the EL model is deployed [22, 2, 46]. We train a bi-encoder baseline EL model (detailed in Section 4.1) on the temporal snapshots from 2014 to 2022 separately and then evaluate EL performance using the test sets of both past and future snapshots.

More specifically, our experiments aim to answer the following research questions: **(Q1)** Does a fixed entity linking (EL) model's performance degrade when applied to newer content? **(Q2)** How does finetuning an EL model on more recent training data affect its performance on both old and newer content? **(Q3)** How does EL performance differ for resolving *new* versus *continual* entities?

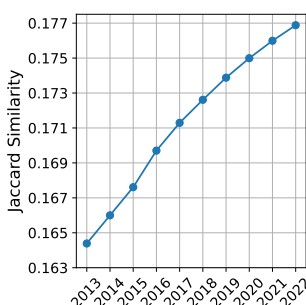 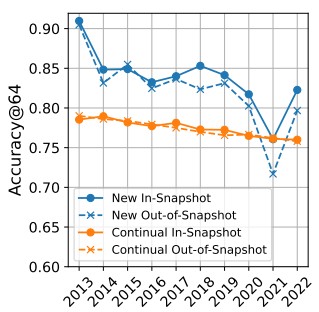 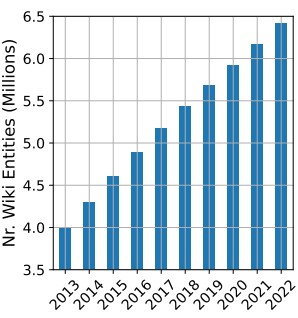

(a) Similarity between candidates returned by the bi-encoder baseline.

(b) Difference in performance between *new* and *continual* entities.

(c) Evolution of the number of entities in the Wikipedia KB.

Figure 4: Statistics related to the analysis of the results (Section 4.2) across the temporal snapshots (x-axis).

## 4.1 Baseline

We experiment with the bi-encoder [47, 16] baseline introduced in the BLINK model [78]. This method independently encodes the mention contexts from the entity descriptions, and then performs the retrieval in a dense space [35] by matching the context of each mention with the closest candidate entities. For the entity description, we concatenate the title to the content of the page describing a particular entity. Both mention context as well as entity descriptions are truncated to 128 BERT tokens as per BLINK model [78]. Similarly to [2, 44], we start from a pre-trained BERT model,[6] which we finetune using our TempEL snapshots' training data — rather than fully re-training the BERT language model on the respective year's full Wikipedia corpus. We leave the latter full-fledged BERT (re-)training approach for future work.

## 4.2 Results and analysis

The results for *continual* and *new* entities are shown in Table 2. The rows thereof represent the snapshots whose train set we used to finetune the bi-encoder model, while the columns indicate the snapshots test data each of the finetuned models was tested on. The used metric is accuracy@64, which amounts to the fraction of anchor mentions in the test set for which the top-64 candidate entity list from the EL model includes the correct target. We observe a consistent temporal decrease in performance for *continual* entities (**Q1**). This is also reflected in Fig. 4b, which illustrates the average temporal degradation across all the finetuned models. We hypothesize that this degradation over time is because, as time evolves, the relative "semantic distance" between the ever growing number of entities shrinks: entities become harder to distinguish from one another. In order to demonstrate this, we calculate the *Jaccard Similarity* between consecutive descriptions of the top 64 candidate entities returned by the bi-encoder. We observe a consistent increase in this similarity metric illustrated in Fig. 4a. This growth in more similar entities is accompanied with a general increase in the number of entities in the Wikipedia KB (see Fig. 4c). Consequently, the model is given an ever-increasing number of candidate target entities, which can potentially impact its performance.

Furthermore, we analyze the impact finetuning on different snapshots has on the performance of the model (**Q2**). To this end, we distinguish between *in-snapshot* and *out-of-snapshot* finetuning setups. In *in-snapshot* setup, the bi-encoder model is finetuned and evaluated on the same snapshot. Conversely, in *out-of-snapshot* setting, the model is evaluated on a different snapshot than the one used for its finetuning. Figure 5a illustrates the difference in performance between the in-snapshot and out-of-snapshot predictions for new and continual entities. We observe a general increase in performance for in-snapshot finetuning with a marginal gain for *continual* entities compared to the *new* ones.[7] This general lower impact of in-snapshot finetuning on *continual* entities, leads us to hypothesize that the actual knowledge needed to disambiguate most of these entities in TempEL changes very little with time. In order to verify this hypothesis, we randomly selected 100 continual

---

[6]We use BERT-large, which is trained on a Wikipedia snapshot from 2018 [34].

[7]We analyze more in detail the difference in performance between *new* and *continual* entities in next paragraphs when addressing (**Q3**).

Table 2: Accuracy@64 for *continual* (top) and *new* (bottom) entities. The intensity of colors is set on a row-by-row basis and indicates whether performance is **better** or **worse** compared to the year the model was finetuned on (i.e., the values that form the white diagonal).

| | Continual Entities | | | | | | | | | |
|---|---|---|---|---|---|---|---|---|---|---|
| Train \ Test | 2013 | 2014 | 2015 | 2016 | 2017 | 2018 | 2019 | 2020 | 2021 | 2022 |
| 2013 | 0.785 | 0.782 | 0.778 | 0.772 | 0.769 | 0.762 | 0.758 | 0.758 | 0.754 | 0.750 |
| 2014 | 0.792 | 0.790 | 0.785 | 0.781 | 0.777 | 0.771 | 0.767 | 0.767 | 0.763 | 0.760 |
| 2015 | 0.786 | 0.784 | 0.782 | 0.777 | 0.773 | 0.769 | 0.765 | 0.764 | 0.760 | 0.757 |
| 2016 | 0.789 | 0.784 | 0.781 | 0.777 | 0.773 | 0.768 | 0.763 | 0.763 | 0.758 | 0.755 |
| 2017 | 0.794 | 0.791 | 0.788 | 0.785 | 0.781 | 0.775 | 0.771 | 0.772 | 0.768 | 0.763 |
| 2018 | 0.791 | 0.788 | 0.786 | 0.782 | 0.778 | 0.773 | 0.769 | 0.769 | 0.764 | 0.760 |
| 2019 | 0.795 | 0.792 | 0.789 | 0.784 | 0.781 | 0.776 | 0.772 | 0.773 | 0.767 | 0.765 |
| 2020 | 0.787 | 0.783 | 0.782 | 0.777 | 0.774 | 0.768 | 0.765 | 0.765 | 0.761 | 0.756 |
| 2021 | 0.788 | 0.785 | 0.782 | 0.777 | 0.773 | 0.769 | 0.764 | 0.764 | 0.761 | 0.757 |
| 2022 | 0.790 | 0.787 | 0.783 | 0.779 | 0.776 | 0.771 | 0.768 | 0.768 | 0.764 | 0.760 |

| | New Entities | | | | | | | | | |
|---|---|---|---|---|---|---|---|---|---|---|
| Train \ Test | 2013 | 2014 | 2015 | 2016 | 2017 | 2018 | 2019 | 2020 | 2021 | 2022 |
| 2013 | 0.910 | 0.819 | 0.853 | 0.826 | 0.841 | 0.812 | 0.819 | 0.791 | 0.688 | 0.774 |
| 2014 | 0.908 | 0.848 | 0.862 | 0.827 | 0.843 | 0.832 | 0.842 | 0.814 | 0.704 | 0.791 |
| 2015 | 0.898 | 0.823 | 0.849 | 0.822 | 0.808 | 0.813 | 0.832 | 0.788 | 0.706 | 0.781 |
| 2016 | 0.897 | 0.832 | 0.862 | 0.832 | 0.839 | 0.823 | 0.823 | 0.802 | 0.718 | 0.791 |
| 2017 | 0.906 | 0.832 | 0.857 | 0.817 | 0.840 | 0.824 | 0.835 | 0.791 | 0.714 | 0.808 |
| 2018 | 0.908 | 0.835 | 0.858 | 0.830 | 0.846 | 0.853 | 0.835 | 0.806 | 0.728 | 0.803 |
| 2019 | 0.910 | 0.842 | 0.853 | 0.821 | 0.842 | 0.843 | 0.841 | 0.810 | 0.734 | 0.799 |
| 2020 | 0.903 | 0.828 | 0.844 | 0.835 | 0.843 | 0.819 | 0.833 | 0.817 | 0.728 | 0.811 |
| 2021 | 0.910 | 0.825 | 0.852 | 0.825 | 0.837 | 0.817 | 0.830 | 0.814 | 0.761 | 0.812 |
| 2022 | 0.905 | 0.846 | 0.852 | 0.820 | 0.830 | 0.830 | 0.832 | 0.808 | 0.732 | 0.823 |

entity-mention pairs, and compared the difference in both mention contexts and entity descriptions between the years 2013 and 2022. We found that in most cases (>95%), while the textual description of the continual entity is changed (supported by Figs. 3a–3b), its meaning remains the same.

Moreover, we address the second part of **Q2** targeting the effect of timespan between the snapshot used for finetuning and the one used for evaluation. To accomplish this, in Fig. 5b we showcase the impact of in-snapshot finetuning relative to the *temporal offset* between the snapshot the model was tested and the snapshot the model was finetuned on. For negative temporal offset,[8] we observe a decrease in the performance difference between in-snapshot and out-of-snapshot setups as the offset approaches to zero. This indicates that the model can benefit more from recent snapshots than from snapshots further in the past. Curiously, we observe a slight increase in performance for out-of-snapshot *continual* entities trained on future snapshots (positive temporal offsets in Fig. 5b). This suggests that the changes in continual entities are *accumulative* in Wikipedia, with later versions of entity descriptions also including the information from the past. For instance, we have observed that for entities describing people, the newly added information on the occupation (e.g., soccer coach) is appended to the occupation description a person had in the past (e.g., soccer player).

Next, we analyze the EL performance on *new* entities and whether they are differently affected than the *continual* ones **(Q3)**. We plot the in-snapshot and out-of-snapshot average temporal change in accuracy@64 scores across all finetuned models for both types of entities in Fig. 4b. We observe that, in general, the performance on *new* entities is superior to that on *continual* ones. Furthermore, as observed above, the performance gain from in-snapshot finetuning on new entities is superior compared to that on continual ones (supported by Fig. 4b and Figs. 5a–5b). This difference suggests that new entities require a higher degree of additional snapshot-specific knowledge to be correctly

---

[8]Evaluation snapshot comes from later time period than the snapshot the model was finetuned on.

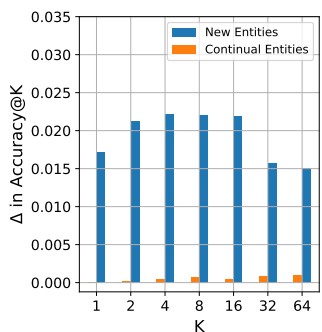
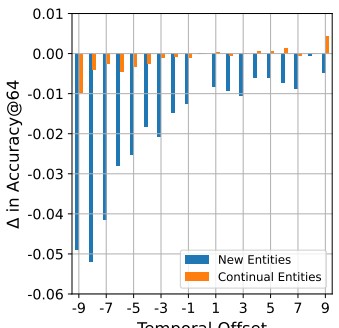
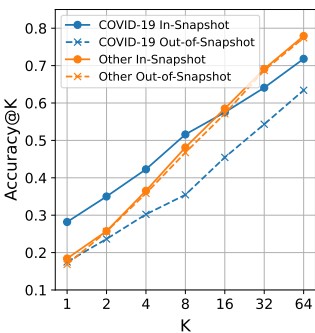

(a) Effect of in-snapshot finetuning (y-axis) across different accuracy thresholds $K$.

(b) In-snapshot finetuning (offset 0) compared to finetuning on past and future snapshots ($-$ and $+$ offsets).

(c) In-snapshot finetuning effect on COVID-19 related and other *new entities* from 2021 snapshot.

Figure 5: Impact of finetuning and evaluating on the same snapshot (*in-snapshot*) compared to finetuning and evaluating on different snapshots (*out-of-snapshot*). We observe: (a) a superior impact of in-snapshot finetuning on *new* entities compared to *continual* ones, (b) a decrease in performance when finetuning on increasingly older spanshots, and (c) dominant effect of in-snapshot finetuning on entities that require fundamentally new knowledge (e.g., COVID-19 related entities).

disambiguated. Additionally, the graph in Fig. 4b reveals that this delta in performance is larger for more recent years (starting from 2018). We hypothesize that this behaviour is due to the fact that the used original BERT model[14] has not been exposed to more recent new entities during pre-training. It also suggests a complementary effect between task-specific finetuning on TempEL dataset and language model pre-training on larger corpora.

Furthermore, to better understand the superior performance on new entities, we manually analyze 100 randomly selected *new* entities from our dataset. We found that a large majority ($\sim$90%) of entities were either events that are recurrent in nature (e.g., "2018 BNP Paribas Open") ($\sim$68%) or extracts of already existing pages ($\sim$22%). We conjecture[9] that these entities require little additional knowledge to be disambiguated, since either they already exist (as part of the content of other entities) or are very similar to already existing entities in Wikipedia. This contrasts sharply with the performance drop observed for *new* entities in the temporal snapshot 2021, as exhibited in both Fig. 4b and Table 2. This decrease is mostly driven by COVID-19 related entities, which constitute 24% of the new entities, which are linked to by 30% of the mentions in this spanshot. The disambiguation of these cases requires completely new and fundamentally different, previously non-existent knowledge. Since this knowledge is not present in the original corpus used to pre-train the BERT encoder nor in any of the previous snapshots, our EL model based on it struggles.

Finally, we analyze the impact of new entities finetuning **(Q2)** on the temporal snapshot 2021, for which our model exhibits the lowest temporal performance driven by COVID-19 disambiguation instances (see above). Figure 5c showcases the impact of in- and out-of-snapshot finetuning on the performance on COVID-19 related entities compared to *other* new entities for different thresholds $K$ of the accuracy@$K$ metric. We observe a large difference in performance (up to 14% accuracy@64 points) between COVID-19 related and the rest of the instances for out-of-snapshot finetuning. This difference is significantly decreased when finetuning on the 2021 snapshot (in-snapshot finetuning), achieving superior accuracy on COVID-19 related entities for lower values of $K$ compared to *other* entities. In contrast, the difference between out- and in-snapshot performance on these non-COVID-19 related entities (*other* entities in Fig. 5c) is marginal. This suggests that in-snapshot finetuning has dominant impact on new entities that require fundamentally new, previously non-existent knowledge in Wikipedia.

---

[9]See Section A.11 of the supplementary material for further details on the performance on these different new entity types.

## 5    Limitations and future work

A number of dataset and model-related aspects were left unexplored in the current work. Our clarifications thereof below may help the community to understand the limitations and potential future research directions to extend our efforts.

**Effect of pre-training on new corpora**    Recent work has demonstrated the benefits of pre-training language models on more recent corpora (e.g., the latest Wikipedia versions) when applied on downstream tasks [2, 44]. We hypothesize that this pre-training may also improve EL performance for our TempEL, especially for *new* entities that require new world knowledge.

**Changes in mention context**    Our work focused mostly on changes in target entities, leaving the effect of changes in mention context on EL performance unexplored. For example, Fig. 3c shows a notable temporal drop in Jaccard vocabulary similarity of the context surrounding mentions. This suggests that mentions, as well as the text surrounding them, are quite volatile and evolve over time, making them an interesting subject for future research.

**Cross-lingual time evolution**    Our dataset is limited to English Wikipedia. Yet, since recent work [5, 11] has shown the benefits of training EL models in a cross-lingual setting, studying cross-lingual temporal evolution of entity linking task may also be an interesting future research direction. Furthermore, it will complement the recent growing interest in creating entity linking datasets for a number of low-resourced languages [24, 52, 6, 67].

## 6    Conclusion

This paper introduced TempEL, a new large-scale temporal entity linking dataset composed of 10 yearly snapshots of Wikipedia target entities linked to by anchor mentions. In our dataset creation pipeline, we put special focus on the quality assurance and future extensibility of TempEL. Furthermore, we established baseline entity linking results across different years, which revealed a noticeable performance deterioration on test data more recent than the training data. We further examined the most challenging cases, suggesting the need for updating the pre-trained language model of our EL model, at least to perform well on newly appearing entities that require new world knowledge (e.g., in case of COVID-19). Finally, we described limitations of our work and discussed potential future research directions.

## Acknowledgments and Disclosure of Funding

Part of the research leading to these results has received funding from (i) the European Union's Horizon 2020 research and innovation programme under grant agreement no. 761488 for the CPN project,[10] (ii) the Flemish Government under the programme "Onderzoeksprogramma Artificiële Intelligentie (AI) Vlaanderen", (iii) the Research Foundation – Flanders grant no. V412922N for Long Stay Abroad at Copenhagen University, and (iv) DFF Sapere Aude grant No 0171-00034B 'Learning to Explain Attitudes on Social Media (EXPANSE)'.

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
