# OpenReview forum: "TempEL: Linking Dynamically Evolving and Newly Emerging Entities"
_NeurIPS.cc/2022/Track/Datasets_and_Benchmarks — NeurIPS 2022 Datasets and Benchmarks _

### Official Review · Reviewer_FKvJ · 2022-07-06
**The paper is not well motivated**

**Rating:** 6
**Confidence:** 4

**Strengths:**

The authors provide a detailed description on the data creation process, particularly on the part cleaning Wikipedia pages.
The quality control in the creation process is discussed in detail.
The dataset is well analyzed through experiments and the authors report a few interesting findings on results.


**Weaknesses:**

The main weakness of the paper is motivation. I do not find the paper well motivated to justify the need of such a dataset. In a typical entity linking setting, we have a document, where a few mentions are detected, then we try to search for candidate entities for each mention and try to link the mentions to their correct entities. The search for candidate entities can be conducted on any KB version.  As long as the KB is updated, then the search is able to return the most up to date entities as candidates.

The second weakness: the dataset does not support collective linking. In a document, typically there are multiple mentions and the confident linking provide additional context for less confident linking, so all the mentions in a document can be collectively linked with confidence. This dataset can not be used to evaluate such models. Further, as the mentions are also from Wikipedia, this dataset does not simulate the linking task in practical setting.

The third weakness is the quality control. It is good that the authors provide the description on how quality control is done. However, due to the quality control, the dataset does not reflect the true performance of a model in real setting (e.g., linking to wikipedia). In real setting, we would not avoid popular entities and it is not wrong to link an entity based on high mention frequency. The control on dataset distribution makes the dataset only for benchmarking purpose, and yet the benchmarking does not reflect the actual performance of a model in real setting.

Thanks for the clarifications from author reply.

**Additional Feedback:**

Overall, I do not find the paper well motivated, and the task setting is fairly different from most entity linking tasks.

Review updated to correct a typo: In real setting, we would **not** avoid popular entities and it is not wrong to link an entity based on high mention frequency.

**Clarity:**

The paper is mostly well written. It would be very helpful if the authors provide some examples in the description. For example, the meaning of mention context. At least some samples of mention and mention context, and entity descriptions. I would consider Figure 1 provides such examples, before reading the supplementary material.

**Correctness:**

The construction of the dataset is sound for the defined objectives in the paper. However, in my understanding, the dataset does not reflect any real problem settings. Hence the performance obtained on this dataset may not truly reflect a model's performance on real settings. An example setting is to link all mentions of names in a news article to Wikipedia entries.

**Documentation:**

The documentation is well written in supplementary material. Again, some of the content currently in supplementary shall be in main text to make the paper self-contained.

**Ethics:**

No concerns.

**Relation To Prior Work:**

Section 2 briefly summarizes related datasets. This dataset is different from existing ones because of different task settings.

**Summary And Contributions:**

The paper presents a dataset called TempEL. TempEL contains Wikipedia snapshots in 10 years and the entity mentions and KB entity descriptions are both drawn from the same Wikipedia text. The authors provide a detailed description on processing Wikipedia pages and sampling entities and mentions.  The authors then conducted an evaluation based on a dense retrieval model.

---

> ### Author Response · Authors · 2022-08-15
> **Author response**
>
> We thank the reviewer for bringing up some of the fundamental aspects behind the creation of the TempEL dataset. The following addresses the raised points.
>
> ---
>
> _The main weakness of the paper is motivation. I do not find the paper well motivated to justify the need of such a dataset. In a typical entity linking setting, we have a document, where a few mentions are detected, then we try to search for candidate entities for each mention and try to link the mentions to their correct entities. The search for candidate entities can be conducted on any KB version. As long as the KB is updated, then the search is able to return the most up to date entities as candidates._
>
> Retraining models on new Knowledge Base (KB) versions is expensive and is a decision that has to be carefully considered. To help with this process, our newly introduced dataset allows to assess the performance degradation associated with linking against “old” KB
> versions, both for entities that evolve over time, as well as for newly appearing entities. This contrasts with the existing entity linking datasets that only consider mentions linked to a single static version of a target KB (e.g., Wikipedia 2010 for CoNLL-AIDA). Concretely, the introduction of TempEL will: (i) enable one to measure the degree to which current models are affected by the inherent time-evolving nature of the entity linking task; and (ii) encourage new research efforts to create robust architectures that are increasingly invariant to temporal changes.
>
> ---
>
> _The second weakness: the dataset does not support collective linking._
>
> As is correctly noted by the reviewer, in the presented dataset we deal with the disambiguation of each of the mentions individually, which is also the case for a substantial set of related works [Provatorova et al., 2021, Logeswaran et al., 2019, Eshel et al., 2017, Onoe and Durrett, 2020]. This format turns the focus on the most challenging cases of entity linking (i.e., filtering out trivial mentions) and suffices to focus on the main research question we study, i.e., the impact of the evolution of KB contents as well as mention contexts over time on EL performance. We view the further complexity of going from per-mention to collective entity linking as complementary line of research, and are confident that the progress made on linking individual mention instances can be incorporated in collective entity linking models.
>
> ---
>
> _Further, as the mentions are also from Wikipedia, this dataset does not simulate the linking task in practical setting._
>
> A number of recent works [Yamada et al., 2022, Barba et al., 2022, Broscheit, 2019, Zhang et al., 2022, Wu et al., 2020] have used Wikipedia links as training signal, achieving state-of-the-art performance on datasets from other domains such as newswire or other Wikis. Additionally, the Wikipedia-based dataset WNED-WIKI [Guo and Barbosa, 2018] has been widely adopted to evaluate the performance of state-of-the-art entity linking models [Barba et al., 2022, Yamada et al., 2022, Xue et al., 2019, De Cao et al., 2021, Fang et al., 2019, Le and Titov, 2019], exhibiting results that are consistent with datasets from other sources. This suggests that entity linking annotations derived from Wikipedia are useful and complementary to the annotations in other domains when used both in training as well as in evaluation settings.
>
> ---
>
> _The third weakness is the quality control. It is good that the authors provide the description on how quality control is done. However, due to the quality control, the dataset does not reflect the true performance of a model in real setting (e.g., linking to Wikipedia). In real setting, we would avoid popular entities and it is not wrong to link an entity based on high mention frequency. The control on dataset distribution makes the dataset only for benchmarking purpose, and yet the benchmarking does not reflect the actual performance of
> a model in real setting._
>
> Here we understand that the reviewer is referring to the dataset distribution paragraph described in our _3.2 Quality Control_ section. We argue that such an adjustment will allow researchers to exclusively study the temporal effect on entity linking performance (i.e., temporal drift in the content of the Wikipedia entities and mentions), without the results being affected by the distinct mention-per-entity distributions across the temporal snapshots. Yet, we understand that some researchers may want to study the effect of temporal change on performance in a more general sense, without such distributional adjustment. Therefore, and as mentioned in _3.2 Quality Control_ section (_Flexibility and extensibility_ paragraph) of our manuscript, we will release our general data generation framework that can be used to
> re-generate the TempEL dataset with different configurations. These will also include the hyperparameters to disable the dataset distribution adjustment during TempEL generation.

---

> > ### Comment · Reviewer_FKvJ · 2022-08-20
> > **Thanks for the reply**
> >
> > Thanks for the replies and I accept some of the clarifications, but not all.
> >
> > Regarding the first point: "the main weakness of the paper is motivation", the reply from authors does not mention to what extent this dataset truly reflects a practical setting in an entity linking task. If we consider that an entity linking task contains two subtasks:(i) candidate entity retrieval, and (ii) candidate ranking. Then candidate retrieval shall work on any KB versions as a typical search. When KB is updated, the retrieval is done on an updated index. The ranking model is to assign a score to each candidate entity, using a learned soring function. The key focus here is to learn the most accurate *scoring function*.
> >
> > Regarding the point on quality control, I do not find the reply reasonable. In an ideal setting, evaluation on a benchmark dataset shall reflect a model's true performance in practical settings. With artificial adjustments on data distribution, the evaluation results no longer reflect model's performance in real life settings. The results obtained on such "exclusively study of the temporal effect on entity linking performance" become less meaningful with respect to a model's true behavior in reality.

---

> > > ### Author Response · Authors · 2022-08-24
> > > **Author Response**
> > >
> > > We thank the reviewer again for valuable comments. Below, we address the raised points.
> > >
> > > _Regarding the first point: "the main weakness of the paper is motivation", the reply from authors does not mention to what extent this dataset truly reflects a practical setting in an entity linking task. If we consider that an entity linking task contains two subtasks:(i) candidate entity retrieval, and (ii) candidate ranking. Then candidate retrieval shall work on any KB versions as a typical search. When KB is updated, the retrieval is done on an updated index. The ranking model is to assign a score to each candidate entity, using a learned soring function. The key focus here is to learn the most accurate scoring function._
> > >
> > > Indeed, accurate candidate retrieval system and ranking functions are important as their errors cascade. Furthermore, in a time-evolving setting, even when entity linkers are up to date with the latest version of the KB (e.g., by updating or adding new entity embeddings), they should also update the models behind the candidate retrievers and candidate rankers. These models rely on training of their respective neural components, which is slow and can be fragile and require careful engineering. Furthermore, these models also have to adapt to the temporal evolution of the anchor mentions (sampled from each of the Wikipedia snapshots in TempEL) that are linked to entities. As such, we expect that TempEL will encourage the development of new architectures that are robust to these temporal changes and, as a consequence, will have longer lifetimes.
> > >
> > > ----
> > >
> > > _Regarding the point on quality control, I do not find the reply reasonable. In an ideal setting, evaluation on a benchmark dataset shall reflect a model's true performance in practical settings. With artificial adjustments on data distribution, the evaluation results no longer reflect model's performance in real life settings. The results obtained on such "exclusively study of the temporal effect on entity linking performance" become less meaningful with respect to a model's true behavior in reality._
> > >
> > > The goal of our work is to investigate the behavior of entity linking systems across time to determine to what degree it is a problem at all. As such, we think it is reasonable to zoom in on data samples that might be sensitive to its effects. Furthermore, as already mentioned in our previously submitted response, we will make this functionality configurable, allowing researchers to explore temporal effects with and without such adjustments in distributions.

---

### Official Review · Reviewer_xkw2 · 2022-07-21
**A useful and well-constructed dataset**

**Rating:** 8
**Confidence:** 4
**Correctness:** Everything seems correct

**Strengths:**

- This is a very nice dataset paper that speaks to a realistic and important application setting (entity linking with continually evolving entities and KBs)
- The continual vs. new entities is a nice framing
- All data is built on wikipedia, which is a great dataset for this setting.
- Provides a strong baseline (BLINK) that is evaluated to measure interesting performance differences between continual and new entities.
- The dataset generation protocol is extremely detailed and clearly spelled out in supplemental materials.
- The dataset is available now.
- Related work is very comprehensive.
- COVID 19 provides a compelling natural experiment for examining temporal entity drift


**Weaknesses:**

- I think the Figure 2 pipeline figure could be clearer. At first glance it doesn't really make sense until read section 3.1 I think adding a "worked" input/output example at key steps of the workflow would help make the process more concrete.

- Some notion of entity types / categories across datasets would be useful for ablation experiments. I'd guess new entities favor certain types (events like "COVID-19_pandemic_in_Portland,_Oregon")

- (Minor) It would be nice to see multiple seeds/replicates for experiments (however since this comes at additional computational resources the absence is reasonable).

**Additional Feedback:**

No additional feedback.

**Clarity:**

- The paper is nicely written and easy to follow. The main area that could be improved is Figure 2, which I think is difficult to parse and doesn't really convey what each step is actually doing in the data workflow.

**Documentation:**

The supplemental material is comprehensive and addresses motivation, maintenance, and other key documentation requirements.

**Ethics:**

No ethical concerns.

**Relation To Prior Work:**

This dataset is a great addition to existing temporal and entity linking datasets. TempEL fills a nice gap in existing datasets by looking at both temporally evolving entities and mentions in a systematic evaluation framework. The manuscript provides a very comprehensive overview of existing entity linking and temporal datasets.

**Summary And Contributions:**

Temporally evolving data streams are present in many application settings. Entity linking is a key application area that is under-explored in the temporal evolving entity setting. This paper presents a dataset, TempEL, that uses time-stratified snapshots of Wikipedia to create a benchmark dataset for evaluating entity linking performance given continually evolving mentions and entities.  The authors build 10 yearly snapshots from Wikipedia (2013-2022), creating snapshots that break down entities into (1) continual entities and (2) new entities.
Using this dataset, the authors evaluate a baseline bi-encoder model (BLINK) to evaluate the impact of temporal drift on entity linking performance.

---

> ### Author Response · Authors · 2022-08-15
> **Author response**
>
> We thank the reviewer for the feedback and comments on our paper.
>
> ---
> _I think the Figure 2 pipeline figure could be clearer. At first glance it doesn’t really make sense until read section 3.1 I think adding a “worked” input/output example at key steps of the workflow would help make the process more concrete._
>
> We are considering adding a more clarifying example for camera ready version.
>
> ---
> _Some notion of entity types / categories across datasets would be useful for ablation experiments. I’d guess new entities favor certain types (events like “COVID-19\_pandemic\_in\_Portland,\_Oregon”)._
>
> This is indeed a promising direction for further ablation experiments. As of now, it is not within the scope of the paper, but we will pick this up in future work, where we plan to combine Wikipedia and Wikidata to get more fine-grained type information. As the dataset entities are linked to Wikidata, this information is accessible for future projects using our dataset.

---

> > ### Comment · Reviewer_xLeS · 2022-08-26
> > **Thank you for the replies.**
> >
> > Thank you for replying ot the comments.
> > I and some of your readers will greatly appreciate the changes and additions in the camera-ready version.

---

### Official Review · Reviewer_xLeS · 2022-07-26
**Well constructed dataset with an interesting and understudied point of view on a useful and requiring task.**

**Rating:** 9
**Confidence:** 3

**Strengths:**

The text is clear, well written and to the point.
The construction of the dataset and the quality check sections are very detailed and show a great effort to provide the cleanest dataset possible.
The benefit of the paper lies in providing a dataset that takes the entity linking evolution in time by intelligently using the Wikimedia system of writing, correction, verification and publication as a controllable proxy for real cases of entity mention.
Entity Linking and Named Entity Recognition are tasks of importance and required steps of many other tasks. An attempt to improve the study of their evolution in time is commendable.

**Weaknesses:**

To serve as a benchmark, the paper could have compared the obtained models (trained on their dataset) with other benchmarks or pre-trained/fine-tuned models from other investigations with their dataset in order to allow a direct comparison and an evident comparison between their work and other SotA work.
The choice of Accuracy@64 metric is not well justified and can be seen as arbitrary and not strict enough without a valid precedent or justification for its use.

**Additional Feedback:**

We recommend the authors use the extra page post review to justify and correct the weaknesses discussed here.

**Clarity:**

The text is clear, well written and to the point. Figures and tables are clear and well explained in the caption.

**Correctness:**

The dataset building is transparent and well explained. The choices taken for its construction are justified and sound.
The dataset is not presented as a benchmark although it could serve as one if it did not lack the required comparison with other related works and models.

**Documentation:**

The authors give plenty detailed descriptions of the data collection and organization. There is no mention of maintenance but their description and source availability allows to easily reconstruct or update the dataset.
We could not find any clear mention on how and if the authors will make the dataset available to the community.

**Ethics:**

There are no to very few ethical concerns since the dataset's source is the open-sourced creative commons dumps of Wikipedia and they describe the use of a pipeline to avoid incorporating highly debatable and doubtful Named entities.

**Relation To Prior Work:**

The paper clearly states that their main contribution lies in the time-evolving aspect of the dataset, which is absent from previous and related works. That statement is true, well exposed and justified.

**Summary And Contributions:**

The authors present TempEL: a new dataset for analyzing and evaluating Entity Linking models on time-evolving cases.
The authors describe in detail the building approaches and quality control precautions to ensure a sound final dataset.
They also provide some basic experiments and results that validate their theory and shows the usability of the dataset.

---

> ### Author Response · Authors · 2022-08-15
> **Author response**
>
> We thank the reviewer for the positive feedback. Below, we address some of the raised points.
>
> ---
> _We could not find any clear mention on how and if the authors will make the dataset available to the community._
>
> As indicated in Section A.2.5 of the supplementary material, the TempEL dataset will be made publicly available in a GitHub repository together with the code to generate it. The baseline code will also be made public on the same repository. Due to its size, the dataset files will be hosted on the cloud server by to Internet Technology and Data Science Lab (IDLab) at Ghent University (https://cloud.ilabt.imec.be/index.php/s/dsqHLKFSq4zJZNk).
>
> ---
> _The choice of Accuracy@64 metric is not well justified and can be seen as arbitrary and not strict enough without a valid precedent or justification for its use._
>
> In principle, Accuracy@64 is the metric originally used in BLINK model (Wu et al., 2020) that we use as a baseline. We updated the supplementary material by adding further results for K ∈ {1, 2, 4, 8, 16, 32, 64} of the accuracy@K metric in Section A.11.
>
> ---
>
> _The dataset is not presented as a benchmark although it could serve as one if it did not lack the required comparison with other related works and models._
>
> We appreciate the perspective of the dataset as a benchmark and agree that it could be used as such in future work. In this work we wanted to introduce the dataset and establish it with the state of the art for entity linking, but are considering creating a task around the dataset as a benchmark for future work.

---

### Official Review · Reviewer_iZsY · 2022-07-28
**Nice dataset**

**Rating:** 7
**Confidence:** 4

**Strengths:**

As far as I can tell, there doesn’t appear to be any other benchmark that enables this type of study. Thus, the contribution is clearly novel.

There’s been a lot of focus on studying the performance of models on knowledge intensive tasks when that knowledge is (1) changing, or (2) less well known (i.e. “tail” entities). I was really glad to see that the authors recognized these directions, and explicitly built test sets to evaluate models along these dimensions.



**Weaknesses:**

No large weaknesses come to mind.

I think additional metrics–beyond just Accuracy@64–would be helpful. I understand why the authors use acc@64 (the Blink paper uses it), but a metric which captures performance more finely (i.e. even accuracy@10) would be interesting to see.

**Additional Feedback:**

N/A

**Clarity:**

The paper is well written. I appreciate the detail the authors devoted to explaining the dataset construction process.

One nit: I’m a bit unclear on where the entity descriptions come from. The paper mentions its anchored in time and that it comes from Wikipedia. Is it the first sentence of the entity’s Wiki page?

**Correctness:**

The construction of the dataset seems sound. I appreciate the care the authors took with regards to the details of the distribution of entities/mentions.

The experiments seem largely correct and well done. My questions are only minor.

Isn’t the upper table (shared entities) in Table 2 a bit misleading? If I train a model on the 2013 split, the number of “unseen” entities relative to the 2013 split is going to be a lot more for the 2018 split than the 2013 split, right? My understanding is that the normalization of unseen entity count is occurring within a year–i.e. the number of unseen entities in the 2018 test relative to 2018 train is the same as the number in the 2013 test relative to the 2013 train. Then, it would make perfect sense why we see the degradation in performance for later years?


Is there a way to quantify the influence of yearly wikipedia pages (e.g. “2018 FIFA WorldCup”) on your metrics? You observe the model seems to have little trouble disambiguating these mentions. I can imagine slicing entities by whether or not their title contains a number. Additional slices (based on entity type) might also provide interesting insights.


**Documentation:**

Yes!

**Ethics:**

No concerns here!

**Relation To Prior Work:**

Yes!

**Summary And Contributions:**

The paper focuses on the problem of temporal shifts for entity linking. In short, as knowledge bases are continually modified–with entities being changed and added–entity linking models can grow stale, having been trained on “older” data. For instance, an entity model train on Wikipedia up til 2019 would have never seen sentences associated with the entity “Coronavirus.” This is a well known problem that has garnered significant interest.

To promote better investigation of this problem, the authors design a benchmark–TempEL–around the idea of temporally shifting entity linking. Specifically, the authors extract 10 yearly snapshots of Wikipedia (2013-22), and for each snapshot, curate a train/val/test split containing entity mentions. This benchmark allows researchers to study how a model trained on one year may generalize to another. The authors describe their process for constructing this dataset, and conduct preliminary experiments to analyze performance.

---

> ### Author Response · Authors · 2022-08-15
> **Author response**
>
> We thank the reviewer for the positive comments and the feedback. Below we address some of the raised points.
>
> ---
> _I think additional metrics–beyond just Accuracy@64–would be helpful. I understand why the authors use acc@64 (the Blink paper uses it), but a metric which captures performance more finely (i.e. even accuracy@10) would be interesting to see._
>
> We updated the supplementary material by adding further results for K ∈ {1, 2, 4, 8, 16, 32, 64} of the accuracy@K metric in Section A.11.
>
> ---
>
> _Isn’t the upper table (shared entities) in Table 2 a bit misleading? If I train a model on the 2013 split, the number of “unseen” entities relative to the 2013 split is going to be a lot more for the 2018 split than the 2013 split, right? My understanding is that the normalization of unseen entity count is occurring within a year–i.e. the number of unseen entities in the 2018 test relative to 2018 train is the same as the number in the 2013 test relative to the 2013 train. Then, it would make perfect sense why we see the degradation in performance for later years?_
>
> In the upper part of Table 2 the caption shared entities referred to continual entities. We fixed this mistake and uploaded the corrected version of the manuscript. Both the number of continual as well as new instances is normalized across snapshots in order to avoid obtaining datasets of different size (e.g., different training set size) across the snapshots. A summary of statistics of TempEL is presented in the Table 1. It is worth noting that “new” entities only consist of entities newly created in the specific year the snapshot is taken from and are not included in “continual” entities subset. Our point was mainly to assess (i) how severe the performance degradation is, and (ii) how it
> differs among entirely “new” entities vs “continual” ones.
>
> ---
>
> _Is there a way to quantify the influence of yearly Wikipedia pages (e.g., “2018 FIFA WorldCup”) on your metrics? You observe the model seems to have little trouble disambiguating these mentions. I can imagine slicing entities by whether or not their title contains a number. Additional slices (based on entity type) might also provide interesting insights._
>
> We have performed an evaluation on the instances that point to new entities that are recurrent (i.e., entities representing events that are repetitive in time such as “2018 FIFA WorldCup”). We use the suggested heuristic of the title containing the year plus some of the keywords of recurrent events such as “league”, “cup”, “election”, etc. On such cases (for snapshot 2021) the accuracy@64 metric is 0.6358 which is significantly above the 0.4265 of COVID-related instances, indicating that such repetitive events are indeed easier to disambiguate than entirely new concepts. We further add Figure 3 in Section A.11 of supplementary material to illustrate the average effect across snapshots of the threshold K of Accuracy@K metric on COVID-related and recurrent new entities.
>
> ---
>
> _I’m a bit unclear on where the entity descriptions come from. The paper mentions its anchored in time and that it comes from Wikipedia. Is it the first sentence of the entity’s Wiki page?_
>
> We concatenate the title to the content of the page describing a particular entity. The resulting token sequence is truncated to the first 128 BERT tokens as per BLINK model [Wu et al., 2020]. We included this clarification in Section 4.1 Baseline of the manuscript.

---

### Official Review · Reviewer_rpfH · 2022-07-28
**Very interesting dataset with funny results.**

**Rating:** 7
**Confidence:** 3
**Correctness:** Yes
**Clarity:** Yes, but it would be better to add mo…

**Strengths:**

This paper focuses on an interesting topic, the dynamic characteristic of entities is important but was less investigated. This paper constructed a large and time award entity linking dataset and examined the model performance in different time settings.

**Weaknesses:**

The whole pipeline of constructing the dataset is not clear. For example, this paper didn't present the number of entities and mentions after each filtering step. Although the supplementary material provides stepwise numbers, a detailed filtering figure with important numbers would make this paper clear.

**Additional Feedback:**

None

**Documentation:**

Yes

**Relation To Prior Work:**

yes

**Summary And Contributions:**

This paper presents an interesting dataset TempEL to explore how do evolving and emerging entities affect the entity linking task.  The proposed dataset includes 10 years snapshot of entities and various experiments were conducted to evaluate the impact of time information on the entity linking task. Experiments show that there is a consistent temporal decrease in performance for continual entities.

One question for the result is in Table 2, why the performance of training/testing in the same year is lower than the performance of training year > testing year? This result is counterintuitive to me.

---

> ### Author Response · Authors · 2022-08-15
> **Author response**
>
> We thank the reviewer for the feedback and acknowledging the value of work. Below we address some of their raised concerns.
>
> ---
>
> _One question for the result is in Table 2, why the performance of training/testing in the same year is lower than the performance of training year > testing year? This result is counterintuitive to me._
>
> This is indeed an interesting observation. The performance on the fine-tuned year tends to indeed be lower than for the models trained on the next years. This underlines our finding that the knowledge to perform the disambiguation changes very little across the snapshots (see lines 253-268). As a result, it is not surprising that for each column in Table 2, the model that performs the best is not trained on the same snapshot on which the test is performed. The observation of the reviewer also suggests that the context around the mentions and/or entity descriptions tends to be more informative (e.g., using linguistically more diverse expressions) in later versions of Wikipedia. This may explain why the models trained on later versions of Wikipedia tend to perform better on the same snapshot.
>
> ---
> _The whole pipeline of constructing the dataset is not clear. For example, this paper didn’t present the number of entities and mentions after each filtering step. Although the supplementary material provides stepwise numbers, a detailed filtering figure with important numbers would make this paper clear._
>
> We have added Figure 1 showcasing the fraction of mentions filtered out by each of the filters in Section A.2.2 of the supplementary material.

---

### Official Review · Reviewer_RJ1h · 2022-07-29
**A useful data set that will become a good benchmark on the task, sadly only in English and needlessly so.**

**Rating:** 8
**Confidence:** 3
**Clarity:** The paper is well written.

**Strengths:**

In my opinion, the main strength of this paper is that it provides a well-structured procedure to create a temporal dataset, taking into account more detailed phenomena than previous data authors. The code is not available, but if it turned out that the code is abstract enough to be able to move beyond Wikipedia, that would make this paper an even stronger submission.

**Weaknesses:**

The main weakness of the paper is that it delivers another Wikipedia-based dataset in a single language, which happens to be English. The English Wikipedia is the most extensively used data source in the most data rich language, what the authors could've done is at least run the method for top 5 Wikipedia languages, or select a lower-resourced language if they only had the budget and means to do one language - even just Belgian and Danish, to have the ability to verify the value of the method outside the most optimistic case of English Wikipedia. The authors mention cross-lingual phenomena as limitations, but truth be told, even a single language low resourced dataset added to this paper would make a great change.


**Additional Feedback:**

It would be very beneficial to the quality of the paper and its impact on the task, if the authors ran their approach on more languages in Wikipedia.

**Correctness:**

The data set is constructed in a well-documented, sound way. Evaluation and experiment design are correct. The supplementary material is extensive.

**Documentation:**

I think that the details on data collection and organizations are extensive and follow ethical and responsible use. The reviewer had access to the data set, not the code. This data set will be a useful benchmark of the task.

**Ethics:**

Following the guidelines:

1-2. The data set surely contains personally identifiable information or information about individuals, as wikipedia contains such information, but as a community we are accepting the ethical standard of wikipedia in this regard.

3. Wikipedia also contains bias, however it also puts significant effort into limiting bias and in this regard, the community also tends to trust wikipedia in that.

4. No human experimentation was performed in this work.

5. Wikipedia has not been discredited by the creators at the time of writing this review.

**Relation To Prior Work:**

The proposed data set introduces mechanisms that control elements, which were not taken into consideration in previous work, such as context availability and dataset balance.

I think it would also be very elegant to cite the efforts of people working on other languages in the same area and task, such as: MobIE by Henning et al., Caillaut et. al for French entity linking, Ogrodniczuk from 2020 for Polish entity linking. The last one was also performed on Wikipedia. Not all of those of course focus on temporal aspects, but remain valuable works in the field - if we want, as a community, to see more work on low-resourced languages we should cite approaches from non-English work in our fields. There's also a PhD thesis by Mendez from 2021 which does a good service to dataset surveying.



**Summary And Contributions:**

The paper introduces an approach to generate a named entity linking dataset with a temporal dimension. It identifies various phenomena related to the problem, and the authors tackle them in the data generation process. As a result the method is used and a dataset is generated and provided, for 10 years of snapshots of English Wikipedia. Additionally, authors establish a baseline performance on this data set.

---

> ### Author Response · Authors · 2022-08-15
> **Author response**
>
> _The main weakness of the paper is that it delivers another Wikipedia-based dataset in a single language, which happens to be English._
>
> Thank you for bringing this point to our attention. We strongly agree with the importance of multilingual datasets. Yet, we leave such multilingual dataset — or indeed, a low-resource language dataset — development for future work. Unquestionably, this would allow to
> study the temporal dependency of entity linking (EL) performance in multilingual/low-resource-language settings. Yet in this first piece of work, our aim was to establish to what extent temporal evolution of Knowledge Bases impacts EL performance. We consider multilinguality to be complementary to our focus of first of all establishing the temporal EL complexities, which we did not want to convolute with multilingual aspects yet. Indeed, multilinguality would raise additional research questions (e.g., should we consider common
> entities across the different Wikis? Should we train on one language and evaluate on others?). Answering these would leave less space for studying temporal effects on entity linking task in depth, for which we are the first to propose a paper. As a conclusion, we consider that a multilingual version of the dataset should be covered in a separate research effort and will explore this direction in the future work.
>
> Having said that, we do believe our data collection scripts should be helpful to replicate the non-trivial dataset collection effort to other languages. Indeed, it already takes a few weeks to train and encode the entity embeddings to get the results for each of the English Wikipedia snapshots in TempEL.
>
> ---
>
> _I think it would also be very elegant to cite the efforts of people working on other languages in the same area and task, such as: MobIE by Henning et al., Caillaut et. al for French entity linking, Ogrodniczuk from 2020 for Polish entity linking. The last one was also performed on Wikipedia. Not all of those of course focus on temporal aspects, but remain valuable works in the field - if we want, as a community, to see more work on low-resourced languages we should cite approaches from non-English work in our fields. There’s also a PhD thesis by Mendez from 2021 which does a good service to dataset surveying._
>
> We thank the reviewer for pointing out the missing related work. We have adapted the paper, adding all of the suggested references.

---

### Meta-Review · Area_Chair_EtYq · 2022-09-02

**Recommendation:** Accept
**Confidence:** 5

**Metareview:**

This paper describes a dataset for entity linking with entities that may change over time (both in how they are referred to as well as their KB description) as well as new, emerging entities. Baselines are evaluated on this dataset.

Reviewers appreciated the general thrust of the work, the resource itself, and the clear presentation. The reviewers were solidly in favor of acceptance overall.

One criticism is the focus on English, which I agree with. There were also criticisms of the Accuracy@64 metric (from BLINK), but these seem fixable in subsequent work if other authors want to use a different evaluation. Finally, reviewer FKvJ raised some issues about both the motivation/task setup as well as some particulars of the dataset construction. However, that reviewer and I found the replies from the authors generally satisfactory. I don't believe there is necessarily a "natural" distribution of entities to link, and choosing to focus on the temporal element is reasonable as a research direction (especially given the growing awareness of temporal divergence in model performance).

---

### Decision · Program_Chairs · 2022-09-16

Accept